# Fast Benchmarking of Asynchronous Multi-Fidelity Optimization on Zero-Cost Benchmarks

**Shuhei Watanabe**[1,*] **Neeratyoy Mallik**[2] **Edward Bergman**[2] **Frank Hutter**[2]

[†]`shuheiwatanabe@preferred.jp`, {`mallik,bergmane,fh`}`@cs.uni-freiburg.de`
[1]Preferred Networks Inc., Japan
[2]University of Freiburg, Department of Computer Science, Germany
[*]This work was done when the author was at the University of Freiburg.

**Abstract**   While deep learning has celebrated many successes, its results often hinge on the meticulous selection of hyperparameters (HPs). However, the time-consuming nature of deep learning training makes HP optimization (HPO) a costly endeavor, slowing down the development of efficient HPO tools. While zero-cost benchmarks, which provide performance and runtime without actual training, offer a solution for non-parallel setups, they fall short in parallel setups as each worker must communicate its queried runtime to return its evaluation in the exact order. This work addresses this challenge by introducing a user-friendly Python package that facilitates efficient parallel HPO with zero-cost benchmarks. Our approach calculates the exact return order based on the information stored in file system, eliminating the need for long waiting times and enabling much faster HPO evaluations. We first verify the correctness of our approach through extensive testing and the experiments with 6 popular HPO libraries show its applicability to diverse libraries and its ability to achieve over 1000x speedup compared to a traditional approach. Our package can be installed via `pip install mfhpo-simulator`.

## 1 Introduction

Hyperparameter (HP) optimization of deep learning (DL) is crucial for strong performance (Zhang et al., 2021; Sukthanker et al., 2022; Wagner et al., 2022) and it surged the research on HP optimization (HPO) of DL. However, due to the heavy computational nature of DL, HPO is often prohibitively expensive and both energy and time costs are not negligible. This is the driving force behind the emergence of zero-cost benchmarks such as tabular and surrogate benchmarks, which enable yielding the (predictive) performance of a specific HP configuration in a small amount of time (Eggensperger et al., 2015, 2021; Arango et al., 2021; Pfisterer et al., 2022; Bansal et al., 2022).

Although these benchmarks effectively reduce the energy usage and the runtime of experiments in many cases, experiments considering runtimes between parallel workers may not be easily benefited as seen in Figure 2b. For example, multi-fidelity optimization (MFO) (Kandasamy et al., 2017) has been actively studied recently due to its computational efficiency (Jamieson and Talwalkar, 2016; Li et al., 2017; Falkner et al., 2018; Awad et al., 2021). To further leverage efficiency, many of these MFO algorithms are designed to maintain their performance under multi-worker asynchronous runs (Li et al., 2020; Falkner et al., 2018; Awad et al., 2021). However, to preserve the return order of each parallel run, a naïve approach involves making each worker wait for the actual DL training to run (see Figure 1 (**Left**)). This time is typically returned as cost of a query by zero-cost benchmarks, leading to significant time and energy waste, as each worker must wait for a potentially long duration.

To address this problem, we introduce algorithms to not wait for large time durations and yet return the correct order of evaluations for each worker via file system synchronization. This is provided as an open-sourced easy-to-use Python wrapper (see Figure 1 (**Right**) for the simplest

Figure 1: The simplest codeblock example of how our wrapper works. **Left**: a codeblock example without our wrapper (naïve simulation). We let each worker call sleep for the time specified by the queried result. This implementation is commonly used to guarantee correctness, as research often requires us to run optimizers from other researchers. **Right**: a codeblock example with our wrapper (multi-core simulation). Users only need to wrap the objective function with our module and remove the line for sleeping. In the end, both codeblocks yield identical results.

codeblock) for existing benchmarking code. Although our wrapper should be applicable to an arbitrary HPO library and yield the correct results universally, it is impossible to perfectly realize it due to different overheads by different optimizers and different multi-core processing methods such as multiprocessing and server-based synchronization. For this reason, we limit our application scope to HPO methods for zero-cost benchmarks with almost no benchmark query overheads. Furthermore, we provide an option to simulate asynchronous optimization over multiple cores only with a single core by making use of the ask-and-tell interface [1].

In our experiments, we first empirically verify our implementation is correct using several edge cases. Then we use various open source software (OSS) HPO libraries such as SMAC3 (Lindauer et al., 2022) and Optuna (Akiba et al., 2019) on zero-cost benchmarks and we compare the changes in the performance based on the number of parallel workers. The experiments demonstrated that our wrapper (see Figure 1 (**Right**)) finishes all the experiments $1.3 \times 10^3$ times faster than the naïve simulation (see Figure 1 (**Left**)). The implementation for the experiments is also publicly available [2].

## 2 Background

In this section, we define our problem setup. Throughout the paper, we assume minimization problems of an objective function [3] $f(x) : \mathcal{X} \to \mathbb{R}$ defined on the search space $\mathcal{X} := \mathcal{X}_1 \times \mathcal{X}_2 \times \cdots \times \mathcal{X}_D$ where $\mathcal{X}_d \subseteq \mathbb{R}$ is the domain of the $d$-th HP. Furthermore, we define the (predictive) *actual* runtime function $\tau(x) : \mathcal{X} \to \mathbb{R}_+$ of the objective function given an HP configuration $x$. Although $f(x)$ and $\tau(x)$ could involve randomness, we only describe the deterministic version for the notational simplicity. In this paper, we use $x^{(n)}$ for the $n$-th sample and $x_n$ for the $n$-th observation and we would like to note that they are different notations. In asynchronous optimization, the sampling order is not necessarily the observation order, as certain evaluations can take longer. For example, if we have two workers and the runtime for the first two samples are $\tau(x^{(1)}) = 200$ and $\tau(x^{(2)}) = 100$, $f(x^{(2)})$ will be observed first, yielding $x_1 = x^{(2)}$ and $x_2 = x^{(1)}$.

### 2.1 Asynchronous Optimization on Zero-Cost Benchmarks

Assume we have a zero-cost benchmark that we can query $f$ and $\tau$ in a negligible amount of time, the $(N+1)$-th HP configuration $x_{N+1}$ is sampled from a policy $\pi(x|\mathcal{D}_N)$ where $\mathcal{D}_N := \{(x_n, f_n)\}_{n=1}^N$ is a set of observations, and we have a set of parallel workers $\{W_p\}_{p=1}^P$ where each worker $W_p : \mathcal{X} \to \mathbb{R}^2$ is a wrapper of $f(x)$ and $\tau(x)$. Let a mapping $i^{(n)} : \mathbb{Z}_+ \to [P]$ be an index specifier of which worker processed the $n$-th sample and $\mathcal{I}_p^{(N)} := \{n \in [N] := \{1, 2, \dots, N\} \mid i^{(n)} = p\}$ be a set of the indices

---

[1] https://optuna.readthedocs.io/en/stable/tutorial/20_recipes/009_ask_and_tell.html

[2] https://github.com/nabenabe0928/mfhpo-simulator-experiments

[3] As mentioned in Appendix B, we can also simulate with multi-objective optimization and constrained optimization.

of samples the $p$-th worker processed. When we define the sampling overhead for the $n$-th sample as $t^{(n)}$, the (simulated) runtime of the $p$-th worker is computed as follows:

$$T_p^{(N)} := \sum_{n \in \mathcal{I}_p^{(N)}} (\tau^{(n)} + t^{(n)}).$$

(1)

Note that $\tau^{(n)}$ includes the benchmark query overhead $t_b$, but we consider it zero, i.e. $t_b = 0$. In turn, the $(N + 1)$-th sample will be processed by the worker that will be free first, and thus the index of the worker for the $(N + 1)$-th sample is specified by $\mathrm{argmin}_{p \in [P]} T_p^{(N)}$. On top of this, each worker needs to free its evaluation when $T_p^{(N)} \le \min_{p' \in [P]} T_{p'}^{(N)} + t_{\mathrm{now}}$ satisfies where $t_{\mathrm{now}}$ is the sampling elapsed time of the incoming sample $\boldsymbol{x}^{(N+1)}$.

The problems of this setting are that (1) the policy $\pi$ is conditioned on $\mathcal{D}_N$, which is why the order of the observations must be preserved, and (2) each worker must wait for the other workers to match the order to be realistic. While an obvious approach is to let each worker wait for the queried runtime $\tau^{(n)}$ as in Figure 1 (**Left**), it is a waste of energy and time. To address this problem, we need a wrapper as in Figure 1 (**Right**).

## 2.2 Related Work

Although there have been many HPO benchmarks invented for MFO such as HPOBench (Eggensperger et al., 2021), NASLib (Mehta et al., 2022), and JAHS-Bench-201 (Bansal et al., 2022), none of them provides a module to allow researchers to simulate runtime internally. We defer the survey by Li and Li (2024) for the details of MFO. Other than HPO benchmarks, many HPO frameworks handling MFO have also been developed so far such as Optuna (Akiba et al., 2019)), SMAC3 (Lindauer et al., 2022), Dragonfly (Kandasamy et al., 2020), and RayTune (Liaw et al., 2018). However, no framework above considers the simulation of runtime. Although HyperTune (Li et al., 2022) and SyneTune (Salinas et al., 2022) are internally simulating the runtime, we cannot simulate optimizers of interest if the optimizers are not introduced in the packages. This restricts researchers in simulating new methods, hindering experimentation and fair comparison. Furthermore, their simulation backend assumes that optimizers take the ask-and-tell interface and it requires the reimplementation of optimizers of interest in their codebase. Since reimplementation is time-consuming and does not guarantee its correctness without tests, it is helpful to have an easy-to-use Python wrapper around existing codes. Note that this work extends previous work (Watanabe, 2023a), by adding the handling of optimizers with non-negligible overhead and the empirical verification of the simulation algorithm.

## 3 Automatic Waiting Time Scheduling Wrapper

As an objective function may take a random seed and fidelity parameters in practice, we denote a set of the arguments for the $n$-th query as $\boldsymbol{a}^{(n)}$. In this section, a *job* means to allocate the $n$-th queried HP configuration $\boldsymbol{x}^{(n)}$ to a free worker and obtain its result $r^{(n)} := (f^{(n)}, \tau^{(n)}) = (f(\boldsymbol{x}^{(n)}|\boldsymbol{a}^{(n)}), \tau(\boldsymbol{x}^{(n)}|\boldsymbol{a}^{(n)}))$. Besides that, we denote the $n$-th chronologically ordered result as $r_n$. Our wrapper outlined in Algorithm 1 is required to satisfy the following conditions:

- The $i$-th result $r_i$ comes earlier than the $j$-th result $r_j$ for all $i < j$,

- The wrapper recognizes each worker and allocates a job to the exact worker even when using multiprocessing (e.g. `joblib` and `dask`) and multithreading (e.g. `concurrent.futures`),

- The evaluation of each job can be resumed in MFO, and

- Each worker needs to be aware of its own sampling overheads.

Note that an example of the restart of evaluation could be when we evaluate DL model instantiated with HP $\boldsymbol{x}$ for 20 epochs and if we want to then evaluate the same HP configuration $\boldsymbol{x}$ for 100

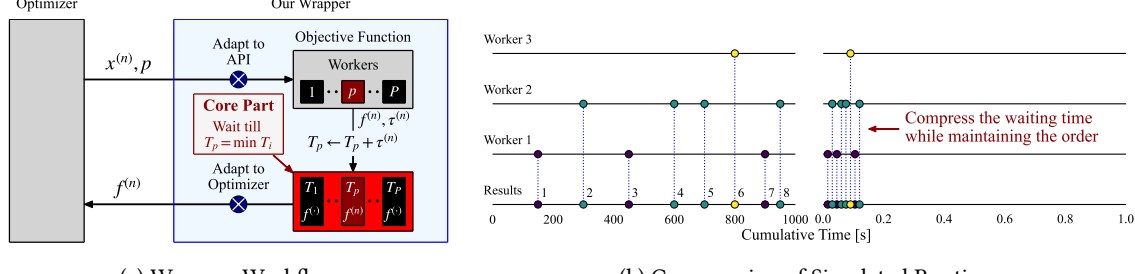

(a) Wrapper Workflow · (b) Compression of Simulated Runtime

Figure 2: The conceptual visualizations of our wrapper. (a) The workflow of our wrapper. The gray parts are provided by users and our package is responsible for the light blue part. The blue circles with the white cross must be modified by users via inheritance to match the signature used in our wrapper. The $p$-th worker receives the $n$-th queried configuration $\boldsymbol{x}^{(n)}$ and stores its result $f^{(n)}, \tau^{(n)}$ in the file system. Our wrapper sorts out the right timing to return the $n$-th queried result $f^{(n)}$ to the optimizer based on the simulated runtime $T_p$. (b) The compression of simulated runtime. Each circle on each line represents the timing when each result was delivered from each worker. **Left**: an example when we naïvely wait for the (actual) runtime $\tau(\boldsymbol{x})$ of each query as reported by the benchmark. **Right**: an example when we use our wrapper to shrink the experiment runtime without losing the exact return order.

epochs, we start the training of this model from the 21st epoch instead of from scratch using the intermediate state. Line 4 checks this condition and Line 5 ensures the intermediate state to restart exists before the evaluation. To achieve these features, we chose to share the required information via the file system and create the following JSON files that map:

- from a thread or process ID of each worker to a worker index $p \in [P]$,

- from a worker index $p \in [P]$ to its timestamp immediately after the worker is freed,

- from a worker index $p \in [P]$ to its (simulated) cumulative runtime $T_p^{(N)}$, and

- from the $n$-th configuration $\boldsymbol{x}^{(n)}$ to a list of intermediate states $s^{(n)} := (\tau^{(n)}, T_{i(n)}^{(n)}, \boldsymbol{a}^{(n)})$.

As our wrapper relies on file system, we need to make sure that multiple workers will not edit the same file at the same time. Furthermore, usecases of our wrapper are not really limited to multi-processing or multithreading that spawns child workers but could be file-based synchronization. Hence, we use fcntl to safely acquire file locks.

   We additionally provide an approach that also extends to the ask-and-tell interface by providing a Single-Core Simulator (SCS) for single-core scenarios (details omitted for brevity). While the Multi-Core Simulator (MCS) wraps optimizers running with $P$ cores or $P$ workers, SCS runs only on a single core and simulates a $P$-worker run. Unlike previous work (Watanabe, 2023a), Algorithm 1 handles expensive optimizers by checking individual workers' wait times during the latest sampling measured by $t_{\text{now}}$ in Line 12. However, this check complicates race conditions, making it hard to guarantee the correctness of implementation. For this reason, empirical verification through edge cases is provided in the next section.

## 4 Empirical Algorithm Verification on Test Cases

In this section, we verify our algorithm using some edge cases. Throughout this section, we use the number of workers $P = 4$. We also note that our wrapper behavior depends only on returned runtime at each iteration in a non-continual setup and it is sufficient to consider only runtime $\tau^{(n)}$ and sampling time $t^{(n)}$ at each iteration. Therefore, we use a so-called fixed-configuration sampler,

**Algorithm 1** Automatic Waiting Time Scheduling Wrapper (see Figure 2a as well)

---

1: **function** WORKER($x^{(N+1)}, a^{(N+1)}$)
2:      Get intermediate state $s^{(N+1)} := (\tau, T, a) = \mathcal{S}.\texttt{get}(x^{(N+1)}, (0, 0, a^{(N+1)}))$
3:      **if** $s^{(N+1)}$ is invalid for $(x^{(N+1)}, a^{(N+1)})$ or $T > T_p^{(N)} + t^{(N+1)}$ **then**
4:          ▷ Cond. 1: The new fidelity input in $a^{(N+1)}$ must be higher than that in $a$ for restart
5:          ▷ Cond. 2: The registration of $s^{(N+1)}$ to $\mathcal{S}$ must happen before the sample of $x^{(N+1)}$
6:          $s^{(N+1)} \leftarrow (0, 0, a^{(N+1)})$
7:      Query the result: $(f^{(N+1)}, \tau^{(N+1)})$
8:      Calibrate runtime for restart: $\tau^{(N+1)} \leftarrow \tau^{(N+1)} - \tau$
9:      $T_{\text{now}} \leftarrow \max(T_{\text{now}}, T_p^{(N)}) + t^{(N+1)}$
10:     $T_p^{(N+1)} \leftarrow T_{\text{now}} + \tau^{(N+1)}, T_{p'}^{(N+1)} \leftarrow T_{p'}^{(N)} \ (p' \neq p)$
11:     ▷ $k$ is the number of results from the other workers that were appended during the wait
12:     ▷ $t_{\text{now}}$ is the sampling elapsed time of the incoming $(N + k + 2)$-th sample $x^{(N+k+2)}$
13:     Wait till $T_p^{(N+1)} = \min_{p' \in [P]} T_{p'}^{(N+k+1)}$ or $T_p^{(N+1)} \leq \min_{p' \in [P]} T_{p'}^{(N+k+1)} + t_{\text{now}}$ satisfies
14:     Record the intermediate state $\mathcal{S}[x^{(N+1)}] = (\tau^{(N+1)}, T_p^{(N+1)}, a^{(N+1)})$
15:     **return** $f^{(N+1)}$

     $\pi$ (an optimizer policy), `get_n_results` (a function that returns the number of recognized results by our wrapper. The results include the ones that have not been reported to the optimizer yet.).
     $\mathcal{D} \leftarrow \emptyset, T_p^{(0)} \leftarrow 0, T_{\text{now}} \leftarrow 0, \mathcal{S} \leftarrow \texttt{dict}()$
16: **while** the budget is left **do**
17:     ▷ This codeblock is run by $P$ different workers in parallel
18:     $N \leftarrow \texttt{get\_n\_results}()$
19:     Get $x^{(N+1)} \sim \pi(\cdot | \mathcal{D})$ and $a^{(N+1)}$ with $t^{(N+1)}$ seconds
20:     $f^{(N+1)} \leftarrow \texttt{worker}(x^{(N+1)}, a^{(N+1)})$
21:     $\mathcal{D} \leftarrow \mathcal{D} \cup \{(x^{(N+1)}, f^{(N+1)})\}$

---

which defines a sequence of HP configurations and their corresponding runtimes at the beginning and samples from the fixed sequence iteratively. More formally, assume we would like to evaluate $N_{\text{all}}$ HP configurations, then the sampler first generates $\{\tau^{(n)}\}_{n=1}^{N_{\text{all}}}$ and one of the free workers receives an HP configuration at the $n$-th sampling that leads to the runtime of $\tau^{(n)}$. Furthermore, we use two different optimizers to simulate the sampling cost:

1. **Expensive Optimizer**: that sleeps for $c(|\mathcal{D}| + 1)$ seconds as a sampling overhead before giving $\tau^{(n)}$ to a worker where $|\mathcal{D}|$ is the size of a set of observations and $c \in \mathbb{R}_+$ is a proportionality constant, and

2. **Cheap Optimizer**: that gives $\tau^{(n)}$ to a worker immediately without a sampling overhead.

In principle, the results of each test case are uniquely determined by a pair of an optimizer and a sequence of runtimes. Hence, we define such pairs at the beginning of each section.

### 4.1 Quantitative Verification on Random Test Cases

We test our algorithm quantitatively using some test cases. The test cases $\{\tau^{(n)}\}_{n=1}^{N_{\text{all}}}$ where $N_{\text{all}} = 100$ for this verification were generated from the following distributions: 1. **Uniform** $\frac{T}{b} \sim \mathcal{U}(0, 2)$, 2. **Exponential** $\frac{T}{b} \sim \text{Exp}(1)$, 3. **Pareto** $\frac{T+1}{b} \sim \mathcal{P}(\alpha = 1)$, and 4. **LogNormal** $\ln \frac{\sqrt{e}T}{b} \sim \mathcal{N}(0, 1)$, where $T$ is the probability variable of the runtime $\tau$ and we used $b = 5$. Each distribution uses the default

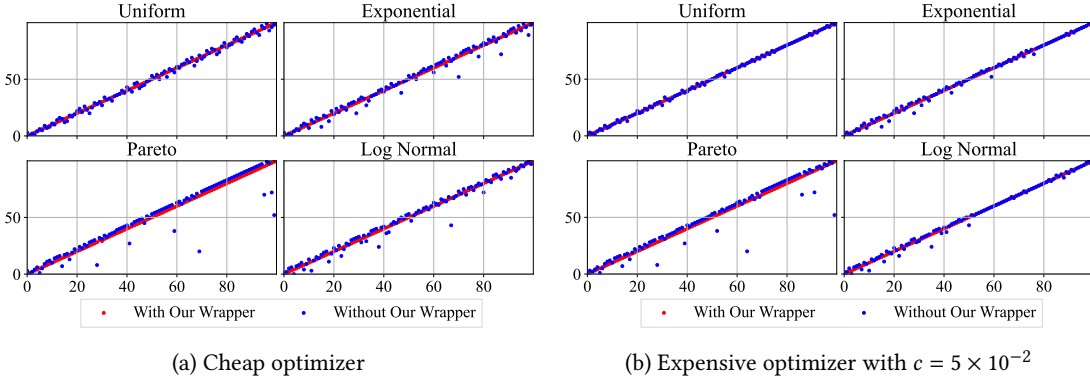

(a) Cheap optimizer

(b) Expensive optimizer with $c = 5 \times 10^{-2}$

Figure 3: The return order verification results. When we use our wrapper, the red dots are obtained. If all the dots are aligned on $y = x$, it implies that the return order in a simulation with our wrapper and that in its naïve simulation perfectly match. As expected, the red dots completely overlap with $y = x$. See the text in "**Checking Return Orders**" for the plot details.

setups of `numpy.random` and the constant number $b$ calibrates the expectation of each distribution except for the Pareto distribution to be 5. Furthermore, we used the cheap optimizer and the expensive optimizer with $c = 5 \times 10^{-2}$. As $N_{\mathrm{all}} = 100$, the worst sampling duration for an expensive optimizer will be 5 seconds. As we can expect longer waiting times for the expensive optimizer, it is more challenging to yield the precise return order and the precise simulated runtime. Hence, these test cases empirically verify our implementation if our wrapper passes every test case.

**Checking Return Orders**. We performed the following procedures to check whether the obtained return orders are correct: (1) run optimizations with the naïve simulation (NS), i.e. Figure 1 (**Left**) and without our wrapper, i.e. Figure 1 (**Right**), (2) define the trajectories for each optimization $\{\tau_n^{\mathrm{NS}}\}_{n=1}^{N_{\mathrm{all}}}$ and $\{\tau_n\}_{n=1}^{N_{\mathrm{all}}}$, (3) sort $\{\tau_n\}_{n=1}^{N_{\mathrm{all}}}$ so that $\{\tau_{i_n}\}_{n=1}^{N_{\mathrm{all}}} = \{\tau_n^{\mathrm{NS}}\}_{n=1}^{N_{\mathrm{all}}}$ holds, and (4) plot $\{(n, i_n)\}_{n=1}^{N_{\mathrm{all}}}$ (see Figure 3). If the simulated return order is correct, the plot $\{(n, i_n)\}_{n=1}^{N_{\mathrm{all}}}$ will look like $y = x$, i.e. $(n, n)$ for all $n \in [N_{\mathrm{all}}]$, and we expect to have such plots for all the experiments. For comparison, we also collect $\{\tau_n\}_{n=1}^{N_{\mathrm{all}}}$ without our wrapper, i.e. Figure 1 (**Left**) *without* `time.sleep` in Line 4. As seen in Figure 3, our wrapper successfully replicates the results obtained by the naïve simulation. The test cases by the Pareto distribution are edge cases because it has a heavy tail and it sometimes generates configurations with very long runtime, leading to blue dots located slightly above the red dots. Although this completely confuses the implementation without our wrapper, our wrapper appropriately handles the edge cases.

**Checking Consistency in Simulated Runtimes**. We check whether the simulated runtimes at each iteration were correctly calculated using the same setups. Figure 4 presents the simulated runtimes for each setup. As can be seen in the figures, our wrapper got a relative error of $1.0 \times 10^{-5} \sim 1.0 \times 10^{-3}$. Since the expectation of runtime is 5 seconds except for the Pareto distribution, the error was approximately $0.05 \sim 5$ milliseconds and this value comes from the query overhead in our wrapper before each sampling. Although the error is sufficiently small, the relative error becomes much smaller when we use more expensive benchmarks that will give a large runtime $\tau^{(n)}$.

## 4.2 Performance Verification on Actual Runtime Reduction

In the previous sections, we verified the correctness of our algorithms and empirically validated our algorithms. In this section, we demonstrate the runtime reduction effect achieved by our wrapper. To test the runtime reduction, we optimized the multi-fidelity 6D Hartmann function [4] (Kandasamy

---

[4] We set the runtime function so that the maximum runtime for one evaluation becomes 1 hour. More precisely, we used $10 \times r(z)$ instead of $r(z)$ in Appendix A.2.

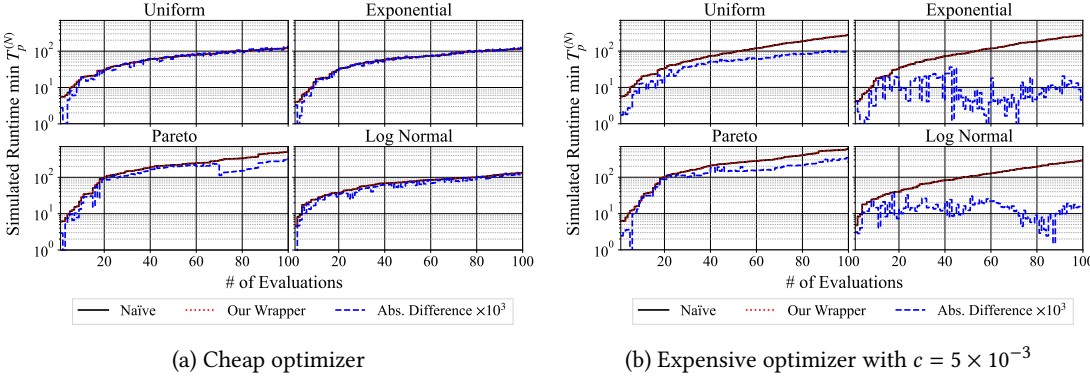

(a) Cheap optimizer

(b) Expensive optimizer with $c = 5 \times 10^{-3}$

Figure 4: The verification of the simulated runtime. The **red** dotted lines show the simulated runtime of our wrapper and the **black** solid lines show the actual runtime of the naïve simulation. The **blue** dotted lines show the absolute difference between the simulated runtime of our wrapper and the actual runtime of the naïve simulation multiplied by 1000 to fit in the same scale as the other lines. The red dotted lines and the black solid lines are expected to completely overlap and the blue lines should exhibit zero ideally. That is, the closer the blue lines to the $x$-axis, the less relative error we have.

et al., 2017) using random search with $P = 4$ workers over 10 different random seeds. In the noisy case, we added a random noise to the objective function. We used both MCS and SCS in this experiment and the naïve simulation. Figure 5 (**Left**) shows that both MCS and SCS perfectly reproduced the results by the naïve simulation while they finished the experiments $6.7 \times 10^3$ times and $1.3 \times 10^5$ times faster, respectively. Note that it is hard to see, but the rightmost curve of Figure 5 (**Left**) has the three lines: (1) Simulated Runtime (MCS), (2) Simulated Runtime (SCS), and (3) Actual Runtime (Naïve), and they completely overlap with each other. SCS is much quicker than MCS because it does not require communication between each worker via the file system. Although MCS could reproduce the results by the naïve simulation even for the noisy case, SCS failed to reproduce the results because the naïve simulation relies on multi-core optimization, while SCS does not use multi-core optimization. This difference affects the random seed effect on the optimizations. However, since SCS still reproduces the results for the deterministic case, it verifies our implementation of SCS. From the results, we can conclude that while SCS is generally quicker because it does not require communication via the file system, it may fail to reproduce the random seed effect. This is because SCS wraps an optimizer by relying on the ask-and-tell interface instead of using the multi-core implementation provided by the optimizer.

## 5 Experiments on Zero-Cost Benchmarks Using Various Open-Sourced HPO Tools

The aim of this section is to show that: (1) our wrapper is applicable to diverse HPO libraries and HPO benchmarks, and that (2) ranking of algorithms varies under benchmarking of parallel setups, making such evaluations necessary. We use random search and TPE (Bergstra et al., 2011; Watanabe, 2023b) from Optuna (Akiba et al., 2019), random forest-based Bayesian optimization (via the MFFacade) from SMAC3 (Lindauer et al., 2022), DEHB (Awad et al., 2021), HyperBand (Li et al., 2017) and BOHB (Falkner et al., 2018) from HpBandSter, NePS [5], and HEBO (Cowen-Rivers et al., 2022) as optimizers. For more details, see Appendix B. Optuna uses multithreading, SMAC3 and DEHB use dask, HpBandSter uses file server-based synchronization, NePS uses file system-based synchronization, and HEBO uses the ask-and-tell interface. In the experiments, we used these optimizers with our wrapper to optimize the MLP benchmark in HPOBench (Eggensperger et al., 2021), HPOLib (Klein and Hutter, 2019), JAHS-Bench-201 (Bansal et al., 2022), LCBench (Zimmer

---

[5]It was under development when we used it and the package is available at `https://github.com/automl/neps/`.

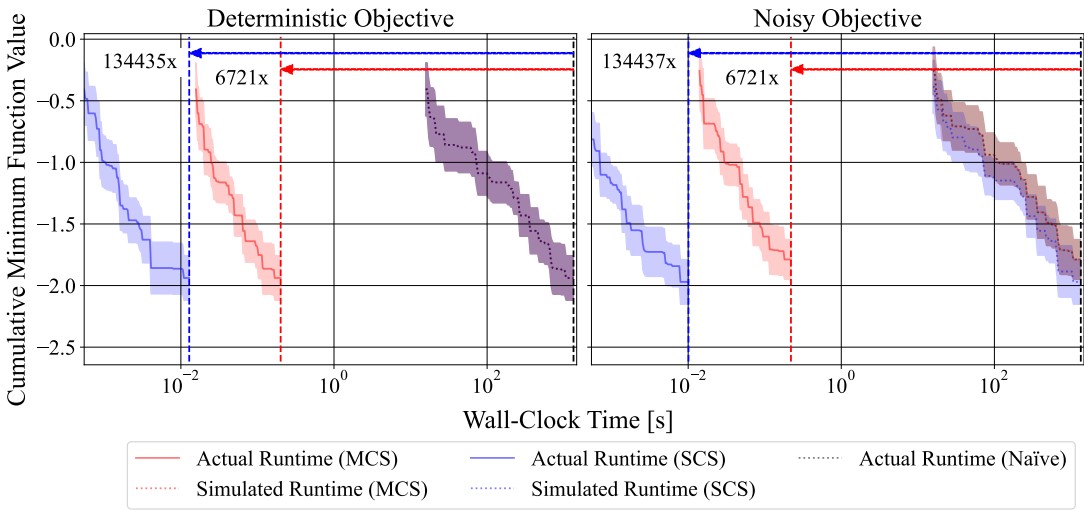

Figure 5: The verification of actual runtime reduction. The $x$-axis shows the wall-clock time and the $y$-axis shows the cumulative minimum objective value during optimizations. Naïve simulation (black dotted line) serves the correct result and the simulated results (red/blue dotted lines) for each algorithm should ideally match the result of the naïve simulation. Actual runtime (red/blue solid lines) shows the runtime reduction compared to the simulated results and it is better if we get the final result as quickly as possible. **Left**: optimization of a deterministic multi-fidelity 6D Hartmann function. The simulated results of our wrapper for both MCS and SCS coincide with the correct result while both of them showed significant speedups. **Right**: optimization of a noisy multi-fidelity 6D Hartmann function. While the simulated result for MCS coincides with the correct result, SCS did not yield the same result. MCS could reproduce the result because MCS still uses the same parallel processing procedure and the only change is to wrap the objective function.

et al., 2021) in YAHPOBench (Pfisterer et al., 2022), and two multi-fidelity benchmark functions proposed by Kandasamy et al. (2017). See Appendix A for more details. We used the number of parallel workers $P \in \{1, 2, 4, 8\}$ over 30 different random seeds for each and $\eta = 3$ for HyperBand-based methods, i.e. the default value of a control parameter of HyperBand that determines the proportion of HP configurations discarded in each round of successive halving (Jamieson and Talwalkar, 2016). The budget for each optimization was fixed to 200 full evaluations and this leads to 450 function calls for HyperBand-based methods with $\eta = 3$. Note that random search and HyperBand used 10 times more budget, i.e. 2000 full evaluations, compared to the others. All the experiments were performed on bwForCluster NEMO, which has 10 cores of Intel(R) Xeon(R) CPU E5-2630 v4 on each computational node, and we used 15GB RAM per worker.

According to Figure 6, while some optimizer pairs such as BOHB and HEBO, and random search and NePS show the same performance statistically over the four different numbers of workers $P \in \{1, 2, 4, 8\}$, DEHB exhibited different performance significance depending on the number of workers. For example, DEHB belongs to the top group with BOHB, TPE, and HEBO for $P = 1$, but it belongs to the bottom group with random search and NePS for $P = 8$. As shown by the red bars, we see statistically significant performance differences between the top groups and the bottom groups. Therefore, this directly indicates that we should study the effect caused by the number of workers $P$ in research. Furthermore, applying our wrapper to the listed optimizers demonstrably accelerated the entire experiment by a factor of $1.3 \times 10^3$ times faster compared to the naïve simulation.

## 6 Broader Impact & Limitations

The primary motivation for this paper is to reduce the runtime of simulations for MFO. As shown in Table 1, our experiments would have taken $5.6 \times 10^{10}$ seconds $\simeq 1.8 \times 10^3$ CPU years with

Table 1: The total actual and simulated runtimes over all the experiments. **Act.**: total actual runtime and **Sim.**: total simulated runtime. × **Fast**: speedup factor of simulation.

| | P = 1 | | | P = 2 | | | P = 4 | | | P = 8 | |
|------|-------|--------|------|-------|--------|------|-------|--------|------|-------|--------|
| Act. | Sim. | × Fast | Act. | Sim. | × Fast | Act. | Sim. | × Fast | Act. | Sim. | × Fast |
| 9.2e+06/ | 3.0e+10/ | **3.3e+03** | 1.1e+07/ | 1.5e+10/ | **1.5e+03** | 1.1e+07/ | 7.7e+09/ | **6.9e+02** | 1.2e+07/ | 3.9e+09/ | **3.2e+02** |

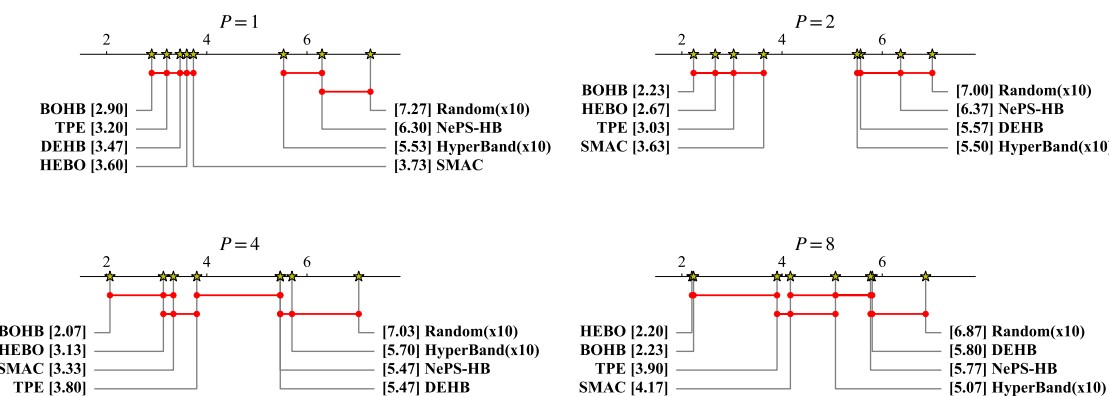

Figure 6: The critical difference diagrams with $1/2^4$ of the runtime budget for random search. "[x.xx]" shows the average rank of each optimizer after using $1/2^4$ of the runtime budget for random search. For example, "BOHB [2.90]" means that BOHB achieved the average rank of 2.90 among all the optimizers after running the specified amount of budget. $P$ indicates the number of workers used and the red bars connect all the optimizers that show no significant performance difference. Note that we used all the results except for JAHS-Bench-201 and LCBench due to the incompatibility between SMAC3, and JAHS-Bench-201 and LCBench.

the naïve simulation. As the TDP of Intel(R) Xeon(R) CPU E5-2630 v4 used in our experiments consumes about 85W and about 350 g of $CO_2$ is produced per 1kWh, the whole experiment would have produced about 9.333 t of $CO_2$ if we estimate a core of the CPU needs 2W in its idole state. It means that our wrapper saved 9.326 t of $CO_2$ production at least. Therefore, researchers can also reduce the similar amount of $CO_2$ for each experiment. The main limitation of our current wrapper is the assumption that none of the workers will not die and any additional workers will not be added after the initialization. Besides that, our package cannot be used on Windows OS because `fcntl` is not supported on Windows.

## 7 Conclusions

In this paper, we presented a simulator for parallel HPO benchmarking runs that maintains the exact order of the observations without waiting for actual runtimes. Our algorithm is available as a Python package that can be plugged into existing code and hardware setups. Although some existing packages internally support a similar mechanism, they are not applicable to multiprocessing or multithreading setups and they cannot be immediately used for newly developed methods. Our package supports such distributed computing setups and researchers can simply wrap their objective functions by our wrapper and directly use their own optimizers. We demonstrated that our package significantly reduces the $CO_2$ production that experiments using zero-cost benchmarks would have caused. Our package and its basic usage description are available at https://github.com/nabenabe0928/mfhpo-simulator.

## Acknowledgments

We acknowledge funding by European Research Council (ERC) Consolidator Grant "Deep Learning 2.0" (grant no. 101045765). Views and opinions expressed are however those of the authors only and do not necessarily reflect those of the European Union or the ERC. Neither the European Union nor the ERC can be held responsible for them. This research was partially supported by TAILOR, a project funded by EU Horizon 2020 research and innovation programme under GA No 952215. We also acknowledge support by the state of Baden-Württemberg through bwHPC and the German Research Foundation (DFG) through grant no INST 39/963-1 FUGG.



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

## A Benchmarks

We first note that since the Branin and the Hartmann functions must be minimized, our functions have different signs from the prior literature that aims to maximize objective functions and when $z = [z_1, z_2, \dots, z_K] \in \mathbb{R}^K$, our examples take $z = [z, z, \dots, z] \in \mathbb{R}^K$. However, if users wish, users can specify $z$ as $z = [z_1, z_2, \dots, z_K]$ from `fidel_dim`.

### A.1 Branin Function

The Branin function is the following $2D$ function that has 3 global minimizers and no local minimizer:

$$f(x_1, x_2) = a(x_2 - bx_1^2 + cx_1 - r)^2 + s(1 - t)\cos x_1 + s \tag{2}$$

where $x \in [-5, 10] \times [0, 15]$, $a = 1$, $b = 5.1/(4\pi^2)$, $c = 5/\pi$, $r = 6$, $s = 10$, and $t = 1/(8\pi)$. The multi-fidelity Branin function was invented by Kandasamy et al. (2020) and it replaces $b, c, t$ with the following $b_z, c_z, t_z$:

$$
\begin{aligned}
b_z &:= b - \delta_b(1 - z_1), \\
c_z &:= c - \delta_c(1 - z_2), \text{ and} \\
t_z &:= t + \delta_t(1 - z_3),
\end{aligned}
\tag{3}
$$

where $z \in [0, 1]^3$, $\delta_b = 10^{-2}$, $\delta_c = 10^{-1}$, and $\delta_t = 5 \times 10^{-3}$. $\delta$. controls the rank correlation between low- and high-fidelities and higher $\delta$. yields less correlation. The runtime function for the multi-fidelity Branin function is computed as [6]:

$$\tau(z) := C(0.05 + 0.95z_1^{3/2}) \tag{4}$$

where $C \in \mathbb{R}_+$ defines the maximum runtime to evaluate $f$.

### A.2 Hartmann Function

The following Hartmann function has 4 local minimizers for the $3D$ case and 6 local minimizers for the $6D$ case:

$$f(x) := -\sum_{i=1}^{4} \alpha_i \exp\left[-\sum_{j=1}^{3} A_{i,j}(x_j - P_{i,j})^2\right] \tag{5}$$

where $\alpha = [1.0, 1.2, 3.0, 3.2]^\top$, $x \in [0, 1]^D$, $A$ for the $3D$ case is

$$
A = \begin{bmatrix}
3 & 10 & 30 \\
0.1 & 10 & 35 \\
3 & 10 & 30 \\
0.1 & 10 & 35
\end{bmatrix},
\tag{6}
$$

$A$ for the $6D$ case is

$$
A = \begin{bmatrix}
10 & 3 & 17 & 3.5 & 1.7 & 8 \\
0.05 & 10 & 17 & 0.1 & 8 & 14 \\
3 & 3.5 & 1.7 & 10 & 17 & 8 \\
17 & 8 & 0.05 & 10 & 0.1 & 14
\end{bmatrix},
\tag{7}
$$

$P$ for the $3D$ case is

$$
P = 10^{-4} \times \begin{bmatrix}
3689 & 1170 & 2673 \\
4699 & 4387 & 7470 \\
1091 & 8732 & 5547 \\
381 & 5743 & 8828
\end{bmatrix},
\tag{8}
$$

---

[6]See the implementation of Kandasamy et al. (2020): `branin_mf.py` at https://github.com/dragonfly/dragonfly/.

and $P$ for the $6D$ case is

$$P = 10^{-4} \times \begin{bmatrix} 1312 & 1696 & 5569 & 124 & 8283 & 5886 \\ 2329 & 4135 & 8307 & 3736 & 1004 & 9991 \\ 2348 & 1451 & 3522 & 2883 & 3047 & 6650 \\ 4047 & 8828 & 8732 & 5743 & 1091 & 381 \end{bmatrix}. \tag{9}$$

The multi-fidelity Hartmann function was invented by Kandasamy et al. (2020) and it replaces $\boldsymbol{\alpha}$ with the following $\boldsymbol{\alpha}_z$:

$$\boldsymbol{\alpha}_z \coloneqq \delta(1 - \boldsymbol{z}) \tag{10}$$

where $\boldsymbol{z} \in [0, 1]^4$ and $\delta = 0.1$ is the factor that controls the rank correlation between low- and high-fidelities. Higher $\delta$ yields less correlation. The runtime function of the multi-fidelity Hartmann function is computed as [7]:

$$\tau(\boldsymbol{z}) = \frac{1}{10} + \frac{9}{10} \frac{z_1 + z_2^3 + z_3 z_4}{3} \tag{11}$$

for the $3D$ case and

$$\tau(\boldsymbol{z}) = \frac{1}{10} + \frac{9}{10} \frac{z_1 + z_2^2 + z_3 + z_4^3}{4} \tag{12}$$

for the $6D$ case where $C \in \mathbb{R}_+$ defines the maximum runtime to evaluate $f$.

### A.3 Zero-Cost Benchmarks

In this paper, we used the MLP benchmark in Table 6 of HPOBench (Eggensperger et al., 2021), HPOlib (Klein and Hutter, 2019), JAHS-Bench-201 (Bansal et al., 2022), and LCBench (Zimmer et al., 2021) in YAHPOBench (Pfisterer et al., 2022).

HPOBench is a collection of tabular, surrogate, and raw benchmarks. In our example, we have the MLP (multi-layer perceptron) benchmark, which is a tabular benchmark, in Table 6 of the HPOBench paper (Eggensperger et al., 2021). This benchmark has 8 classification tasks and provides the validation accuracy, runtime, F1 score, and precision for each configuration at epochs of $\{3, 9, 27, 81, 243\}$. The search space of MLP benchmark in HPOBench is provided in Table 2.

HPOlib is a tabular benchmark for neural networks on regression tasks (Slice Localization, Naval Propulsion, Protein Structure, and Parkinsons Telemonitoring). This benchmark has 4 regression tasks and provides the number of parameters, runtime, and training and validation mean squared error (MSE) for each configuration at each epoch. The search space of HPOlib is provided in Table 3.

JAHS-Bench-201 is an XGBoost surrogate benchmark for neural networks on image classification tasks (CIFAR10, Fashion-MNIST, and Colorectal Histology). This benchmark has 3 image classification tasks and provides FLOPS, latency, runtime, architecture size in megabytes, test accuracy, training accuracy, and validation accuracy for each configuration with two fidelity parameters: image resolution and epoch. The search space of JAHS-Bench-201 is provided in Table 4.

LCBench is a random-forest surrogate benchmark for neural networks on OpenML datasets. This benchmark has 34 tasks and provides training/test/validation accuracy, losses, balanced accuracy, and runtime at each epoch. The search space of HPOlib is provided in Table 5.

## B Optimizers

In our package, we show examples using BOHB (Falkner et al., 2018), DEHB (Awad et al., 2021), SMAC3 (Lindauer et al., 2022), and NePS [8]. BOHB is a combination of HyperBand (Li et al., 2017) and

---

[7]See the implementation of Kandasamy et al. (2020): `hartmann3_2_mf.py` for the $3D$ case and `hartmann6_4_mf.py` for the $6D$ case at https://github.com/dragonfly/dragonfly/.

[8]https://github.com/automl/neps/

Table 2: The search space of the MLP benchmark in HPOBench (5 discrete + 1 fidelity parameters). Note that we have 2 fidelity parameters only for the raw benchmark. Each benchmark has performance metrics of 30000 possible configurations with 5 random seeds.

| Hyperparameter | Choices |
|---|---|
| L2 regularization | $[10^{-8}, 1.0]$ with 10 evenly distributed grids |
| Batch size | $[4, 256]$ with 10 evenly distributed grids |
| Initial learning rate | $[10^{-5}, 1.0]$ with 10 evenly distributed grids |
| Width | $[16, 1024]$ with 10 evenly distributed grids |
| Depth | $\{1, 2, 3\}$ |
| Epoch (**Fidelity**) | $\{3, 9, 27, 81, 243\}$ |

Table 3: The search space of HPOlib (6 discrete + 3 categorical + 1 fidelity parameters). Each benchmark has performance metrics of 62208 possible configurations with 4 random seeds.

| Hyperparameter | Choices |
|---|---|
| Batch size | $\{2^3, 2^4, 2^5, 2^6\}$ |
| Initial learning rate | $\{5 \times 10^{-4}, 10^{-3}, 5 \times 10^{-3}, 10^{-2}, 5 \times 10^{-2}, 10^{-1}\}$ |
| Number of units {1,2} | $\{2^4, 2^5, 2^6, 2^7, 2^8, 2^9\}$ |
| Dropout rate {1,2} | $\{0.0, 0.3, 0.6\}$ |
| Learning rate scheduler | {cosine, constant} |
| Activation function {1,2} | {relu, tanh} |
| Epoch (**Fidelity**) | $[1, 100]$ |

tree-structured Parzen estimator (Bergstra et al., 2011; Watanabe, 2023b). DEHB is a combination of HyperBand and differential evolution. We note that DEHB does not natively support restarting of models, which we believe contributes to it subpar performance. SMAC3 is an HPO framework. SMAC3 supports various Bayesian optimization algorithms and uses different strategies for different scenarios. The default strategies for MFO is the random forest-based Bayesian optimization and HyperBand. NePS is another HPO framework jointly with neural architecture search. When we used NePS, this package was still under developed and we used HyperBand, which was the default algorithm at the time. Although we focused on multi-fidelity optimization in this paper, our wrapper is applicable to multi-objective optimization and constrained optimization. We give examples for these setups using MO-TPE (Ozaki et al., 2020, 2022) and c-TPE (Watanabe and Hutter, 2022, 2023) at `https://github.com/nabenabe0928/mfhpo-simulator/blob/main/examples/minimal/optuna_mo_ctpe.py`.

Table 4: The search space of JAHS-Bench-201 (2 continuous + 2 discrete + 8 categorical + 2 fidelity parameters). JAHS-Bench-201 is an XGBoost surrogate benchmark and the outputs are deterministic.

| Hyperparameter | Range or choices |
|---|---|
| Learning rate | $[10^{-3}, 1]$ |
| L2 regularization | $[10^{-5}, 10^{-2}]$ |
| Activation function | {ReLU, Hardswish, Mish} |
| Trivial augment (Müller and Hutter (2021)) | {True, False} |
| Depth multiplier | $\{1, 2, 3\}$ |
| Width multiplier | $\{2^2, 2^3, 2^4\}$ |
| Cell search space (NAS-Bench-201 (Dong and Yang (2020)), Edge $1 - 6$) | {none, avg-pool-3x3, bn-conv-1x1, bn-conv-3x3, skip-connection} |
| Epoch (**Fidelity**) | $[1, 200]$ |
| Resolution (**Fidelity**) | $[0.0, 1.0]$ |

Table 5: The search space of LCBench (3 discrete + 4 continuous + 1 fidelity parameters). Although the original LCBench is a collection of 2000 random configurations, YAHPOBench created random-forest surrogates over the 2000 observations. Users can choose deterministic or non-deterministic outputs.

| Hyperparameter | Choices |
|---|---|
| Batch size | $[2^4, 2^9]$ |
| Max number of units | $[2^6, 2^{10}]$ |
| Number of layers | $[1, 5]$ |
| Initial learning rate | $[10^{-4}, 10^{-1}]$ |
| L2 regularization | $[10^{-5}, 10^{-1}]$ |
| Max dropout rate | $[0.0, 1.0]$ |
| Momentum | $[0.1, 0.99]$ |
| Epoch (**Fidelity**) | $[1, 52]$ |

