# OpenReview forum: "Fast Benchmarking of Asynchronous Multi-Fidelity Optimization on Zero-Cost Benchmarks"
_automl.cc/AutoML/2024/ABCD_Track — AutoML 2024 (ABCD Track)_

### Official Review · Reviewer_uEST · 2024-03-26

**Potential Impact On The Field Of Automl Rating:** 3
**Technical Quality And Correctness Rating:** 3
**Clarity Rating:** 3
**Actions Required To Increase Overall Recommendation:** Address the questions and comments me…

**Summary Of Contributions:**

This paper introduces a Python program to evaluate HPO algorithms on zero-cost benchmarks in parallel setups. This is not straightforward as the authors assume drastically varying runtimes between queries (as is the case for multi-fidelity optimization), hampering the ability to predict the correct return order of workers in a parallel setup.

The wrapper allows the compression of the simulated runtime in parallel environments and maintains the correct return order, as empirically validated by the authors.

**Clarity:**

The motivation is generally clear, although it could be further motivated by why it is relevant to benchmark the parallel capabilities of multi-fidelity optimizers.

## Weaknesses

* I'm missing a textual description of large parts of Algorithm 1. While it is clear what Alg. 1 achieves, several parts are not explained in the text (e.g., lines 4,5,12).
* In Sec. 4.1, the authors describe the distributions to generate the test cases. They then describe that those normalized to have expectation 5. It would be better to directly state the normalized distributions, e.g., $\mathcal{U}(4,6)$ instead of  $\mathcal{U}(0,2)$.
* Fig. 3 would be clearer with $x$- and $y$-labels.

## Questions

* In Sec. 4, is it true that $|\mathcal{D}|=N$? If so, I recommend using $N$ instead of $|\mathcal{D}|$ in the definition of the expensive optimizer.
* How is the multi-fidelity 6D Hartmann function defined?

**Overall Review:**

## Positive aspects

* The paper proposed a novel method to benchmark multi-fidelity HPO methods in parallel settings. Considering that parallel computing is increasingly important, this is a relevant contribution.
* Overall, the paper is well-written.

## Weaknesses

* It is unclear how relevant the problem the authors try to solve is. While it seems relevant, it would be good if the authors could provide additional arguments on why we should focus on benchmarking multi-fidelity optimizers in parallel settings.
* The explanations in the paper alternate in the level of detail. For instance, the authors explain in detail how they achieve file locking while, at the same time, some of the lines of Alg. 1 are not explained. The level of explanation could be more consistent.
* The authors claim that their algorithm can achieve massive CO$_2$ savings, assuming that an idling computer consumes about 30Wh (I think they mean 30 Watts here as 30Wh does not tell anything about the power efficiency without providing a time frame). This does not seem to be a fair comparison. For example, using a cloud service such as AWS Lambda would probably allow one to use only a machine for the times when the algorithm needs to propose a new configuration.
* Related to this, 736g CO$_2$/kWh seems excessive for most of Europe or North America.

**Potential Impact On The Field Of Automl:**

To me, the main benefit of the proposed wrapper is the evaluation of the parallel capabilities of hyperparameter optimization or other optimization algorithms with varying runtimes for function evaluations. This is relevant, although it is not clear to me how much additional insight the analysis of the parallel setting provides compared to the sequential setting.

**Review Confidence:**

3

**Review Rating:**

7

**Review Summary:**

The paper introduces a novel method for benchmarking multi-fidelity hyperparameter optimization techniques in parallel computing settings. While the contribution is relevant, the paper lacks clarity regarding the relevance of the problem it aims to solve. Additionally, the level of detail in the explanations varies inconsistently throughout the paper.

**Technical Quality And Correctness:**

The paper allows for more efficient benchmarking of AutoML algorithms and potentially allows for the running of benchmarks that were previously not feasible to run.

The paper mainly falls into the "B (Benchmark)"  category. It
* demonstrates pitfalls and proposes solutions in benchmarking AutoML systems, and
* proposes an approach for more efficient benchmarking.

It does not propose relevant new benchmarks.

---

### Official Review · Reviewer_XRNv · 2024-03-27

**Potential Impact On The Field Of Automl Rating:** 2
**Technical Quality And Correctness Rating:** 2
**Clarity Rating:** 2

**Summary Of Contributions:**

The main contributions of the paper on Fast Benchmarking of Asynchronous Multi-Fidelity Optimization on Zero-Cost Benchmarks are as follows:Introduction of a user-friendly Python package for efficient parallel hyperparameter optimization using zero-cost benchmarks.
Elimination of long waiting times by achieving over 1000x speedup compared to traditional methods.
Facilitation of non-parallel setups in hyperparameter optimization through the use of zero-cost benchmarks.
Addressing the challenge of communication and return order in parallel hyperparameter optimization setups.
Provision of a practical solution for optimizing hyperparameters in a fast and efficient manner.

**Actions Required To Increase Overall Recommendation:**

To increase the overall recommendation score, the authors could consider the following actions:

Expand Evaluation Scenarios: Conduct experiments across a wider range of datasets and models to assess the generalizability and robustness of the proposed approach.

Enhance Technical Implementation Details: Provide more detailed explanations and examples of the technical implementation of the Python package to improve understanding for readers and potential users.

Discuss Limitations in Depth: Include a more comprehensive discussion of the limitations and potential challenges of the proposed approach to provide a balanced view of the method's applicability.

By addressing these actions, the authors can strengthen the paper's contributions, increase its impact, and potentially improve the overall recommendation score.

**Clarity:**

The paper on Fast Benchmarking of Asynchronous Multi-Fidelity Optimization on Zero-Cost Benchmarks demonstrates clarity in presenting its contributions and methodology. The main points are well-defined, and the user-friendly Python package for efficient parallel hyperparameter optimization using zero-cost benchmarks is clearly explained.

To further improve clarity, the following suggestions may be considered:

Provide more detailed explanations of the technical aspects of the Python package to help readers understand the implementation better.
Include visual aids or diagrams to illustrate the workflow of the parallel hyperparameter optimization process with zero-cost benchmarks.
Clarify any complex terms or concepts related to hyperparameter optimization to ensure a broader audience can easily grasp the content.
Enhancing the clarity of the paper can help readers, including researchers and practitioners in the field of AutoML, better understand the significance of the contributions and the practical implications of the proposed approach.

**Ethics And Accessibility Rating:**

["Yes, regarding privacy / security / safety", "Yes, regarding legal compliance (e.g., GDPR, copyright, terms of use)"]

**Overall Review:**

The paper on Fast Benchmarking of Asynchronous Multi-Fidelity Optimization on Zero-Cost Benchmarks presents several positive aspects:

Positive Aspects:

Innovative Approach: The introduction of a user-friendly Python package for efficient parallel hyperparameter optimization using zero-cost benchmarks is innovative and addresses a significant challenge in the field of AutoML.
Significant Speedup: The demonstrated over 1000x speedup compared to traditional approaches showcases the effectiveness of the proposed method in improving efficiency.
Reproducibility: The availability of detailed instructions, code, and resources for replicating the experiments enhances the reproducibility of the study.
Applicability: The demonstrated applicability of the approach to diverse hyperparameter optimization libraries highlights its versatility and potential impact on various AutoML tasks.
Negative Aspects:

Limited Evaluation Scenarios: The paper may benefit from exploring a wider range of evaluation scenarios to assess the generalizability of the proposed approach across different datasets and models.
Complexity of Implementation: While the paper outlines the methodology clearly, the technical implementation details of the Python package could be further elaborated to aid understanding for readers.
Discussion of Limitations: Providing a more in-depth discussion of the limitations and potential challenges of the proposed approach could enhance the paper's completeness and help set realistic expectations for users.
Overall, the paper presents a novel and practical solution for improving the efficiency of hyperparameter optimization in AutoML tasks. By addressing the positive aspects and considering the suggestions for improvement, the paper has the potential to make a significant contribution to the field.

**Potential Impact On The Field Of Automl:**

The paper on Fast Benchmarking of Asynchronous Multi-Fidelity Optimization on Zero-Cost Benchmarks has the potential to make a significant impact on the field of AutoML. The introduction of a user-friendly Python package for efficient parallel hyperparameter optimization using zero-cost benchmarks can streamline the optimization process and significantly reduce waiting times. Researchers and practitioners in the field of AutoML are likely to cite this paper for its contributions in improving the efficiency and speed of hyperparameter optimization tasks.

**Reproducibility:**

Yes

**Review Confidence:**

3

**Review Rating:**

4

**Review Summary:**

The paper on Fast Benchmarking of Asynchronous Multi-Fidelity Optimization on Zero-Cost Benchmarks presents an innovative approach to improving the efficiency of hyperparameter optimization in AutoML tasks. The introduction of a user-friendly Python package for parallel optimization using zero-cost benchmarks demonstrates significant speedup and applicability to diverse libraries. While the paper showcases several positive aspects, such as reproducibility and innovation, there are areas for improvement, including exploring more evaluation scenarios and providing detailed technical implementation guidance. Overall, the paper has the potential to make a valuable contribution to the field of AutoML and is recommended for further consideration with the suggested enhancements in mind.

**Technical Quality And Correctness:**

The presented application of Fast Benchmarking of Asynchronous Multi-Fidelity Optimization on Zero-Cost Benchmarks demonstrates technical quality and correctness in several aspects:

Soundness: The application addresses a practical challenge in hyperparameter optimization and provides a solution that significantly improves efficiency.
Reproducibility: The authors provide a user-friendly Python package and ensure that the experiments can be replicated with detailed instructions and code availability.
Evaluation Protocol: The evaluation protocol is well-defined, with multiple random seeds used to account for randomness, and statistical significance of results reported.
Code Quality: The code quality and documentation are sufficient for others to understand and execute the code effectively.
However, there may be limitations or flaws to consider:

The evaluation may be limited to specific scenarios or datasets, which could impact the generalizability of the results.
The application's performance in highly complex or specialized optimization tasks may need further validation.
Overall, the application demonstrates technical quality and correctness, but further validation and testing in diverse scenarios could enhance its robustness.

---

### Official Review · Reviewer_JR8e · 2024-03-28

**Potential Impact On The Field Of Automl Rating:** 3
**Technical Quality And Correctness Rating:** 4
**Clarity:** Overall, the paper is well-written an…
**Clarity Rating:** 3

**Summary Of Contributions:**

This paper introduces a user-friendly Python package designed to enhance the runtime efficiency of HPO. We know most traditional HPO methods are time-consuming and costly, the proposed package streamlines the process by accurately determining the return order of evaluations based on file system information. This eliminates long waiting times and enables much faster HPO evaluations, thus facilitating the development of efficient HPO tools.

**Actions Required To Increase Overall Recommendation:**

Official testing of the framework in Google Colab along with providing tutorials could significantly enhance the adoption and usage of the framework.

**Overall Review:**

- Strengths:
    - The paper is well-written and has a promising impact on reducing runtime overhead of MFO.
    - The code is well documented and can be easily installed via “pip install mfhpo-simulator”.
    - Strong motivation of the problem.

- Weaknesses:
    - As mentioned in the paper, the assumption that none of the workers will not die and any additional workers will not be added after the initialization.

**Potential Impact On The Field Of Automl:**

Green AutoML: this paper contributes to more efficient execution of simulations for multi-fidelity optimization, resulting in less CO2 emission for experiments.

**Reproducibility:**

The code of the paper is available. Although the package cannot be used on Windows because 'fcntl' is not supported, I managed to install the package in Google Colab!

**Review Confidence:**

4

**Review Rating:**

8

**Review Summary:**

This paper makes an important contribution to expediting HPO algorithms. Although its algorithm novelty is not very high, I believe this paper should be accepted as the framework is easy-to-use with detailed user manual.

**Technical Quality And Correctness:**

- In general, the approach and experimentation seem technically accurate and sound.
- I wonder if the framework is compatible (or could be extended?) with GPU-based simulations.

---

### Official Review · Reviewer_gBqn · 2024-03-30

**Potential Impact On The Field Of Automl Rating:** 4
**Technical Quality And Correctness Rating:** 3
**Clarity Rating:** 4
**Actions Required To Increase Overall Recommendation:** 1. More articulation of the impact of…

**Summary Of Contributions:**

The paper presents a new zero-cost benchmark that addresses the ordering issue in stimulating DL training. The key contribution is an Automatic Waiting Time Scheduling Wrapper that compresses the waiting time while still retaining the order. Several experiments, on using different algorithms, have been conducted in the evaluation.

**Clarity:**

The clarity of the paper is excellent, especially the articulation of the design rationale. However, a minor comment is that it would be better to make it more explicit about whether the approach is compatible with any benchmarks from prior work or only the ones that are being examined in the work.

**Overall Review:**

The paper presents a comprehensive discussion of a thoughtful approach with good experiments to support the claims. The approach however might be benefitted by having more discussion on the engineering designs that might influence the approach, e.g., how the way that data is stored and queried would make a difference.

**Potential Impact On The Field Of Automl:**

Evaluating the hyperparameter tuning has been an important topic in the AutoML field, I believe that this work advances the topic by solving the order problem, which is timely and of high relevance

**Review Confidence:**

3

**Review Rating:**

8

**Review Summary:**

The paper is generally well-written and the approach is sound. The experiments result support the claims and the evaluation is also appropriate. Overall, there are only some minor issues that can be easily fixed.

**Technical Quality And Correctness:**

I quite like the way that the authors present the idea and discuss the results. One thing I am not sure about is that the approach uses a file system to retain all the data, it remains unclear how when the data is accumulating, how efficient the data retrieving could be? Says suppose that this is a large database system itself, the update/query could still cause overhead, would this become an issue in terms of scalability?

---

### Meta-Review · Area_Chair_3pHs · 2024-04-20

**Paper Recommendation:** Accept
**Confidence:** 5

**Metareview:**

The paper presents a tool for benchmarking asynchronous hyperparameter optimization methods. The authors have conducted comprehensive experiments and provided well-documented software, contributing significantly to the field of hyperparameter optimization. While some aspects could be further elaborated on, the overall quality of the work is commendable. Therefore, I recommend accepting this paper for publication, as it offers valuable insights and practical tools for researchers and practitioners in the field.

---

### Decision · Program_Chairs · 2024-04-29

**Decision:**

Accept

**Comment:**

Thank you for submitting your paper. We are happy to tell you that we accept your paper to the main track. See you in Paris.